# A surface pocket in the cytoplasmic domain of the herpes simplex virus fusogen gB controls membrane fusion

Zemplen Pataki[1,2], Erin K. Sanders[3], Ekaterina E. Heldwein[1,2,3]*

**1** Department of Molecular Biology and Microbiology, Tufts University School of Medicine, Boston, Massachusetts, United States of America, **2** Graduate Program in Molecular Microbiology, Graduate School of Biomedical Sciences, Tufts University School of Medicine, Boston, Massachusetts, United States of America, **3** Graduate Program in Cellular, Molecular, and Developmental Biology, Graduate School of Biomedical Sciences, Tufts University School of Medicine, Boston, Massachusetts, United States of America

* katya.heldwein@tufts.edu

**Data Availability Statement:** All relevant data are within the paper and its Supporting Information files.

**Funding:** EEH was supported by grant R01AI164698 from the National Institutes of Health

## Abstract

Membrane fusion during the entry of herpesviruses is carried out by the viral fusogen gB that is activated by its partner protein gH in some manner. The fusogenic activity of gB is controlled by its cytoplasmic (or intraviral) domain ($gB_{CTD}$) and, according to the current model, the $gB_{CTD}$ is a trimeric, inhibitory clamp that restrains gB in the prefusion conformation. But how the $gB_{CTD}$ clamp is released by gH is unclear. Here, we identified two new regulatory elements within gB and gH from the prototypical herpes simplex virus 1: a surface pocket within the $gB_{CTD}$ and residue V831 within the gH cytoplasmic tail. Mutagenesis and structural modeling suggest that gH V831 interacts with the gB pocket. The gB pocket is located above the interface between adjacent protomers, and we hypothesize that insertion of the gH V831 wedge into the pocket serves to push the protomers apart, which releases the inhibitory clamp. In this manner, gH activates the fusogenic activity of gB. Both gB and gH are conserved across all herpesviruses, and this activation mechanism could be used by other gB homologs. Our proposed mechanism emphasizes a central role for the cytoplasmic regions in regulating the activity of a viral fusogen.

## Author summary

Herpes simplex virus 1 (HSV-1) establishes lifelong infections in over a half of people and causes diseases ranging from oral or genital sores to blindness and brain inflammation. No vaccines or curative treatments are currently available. To infect cells, HSV-1 must first penetrate them by merging its lipid envelope with the membrane of the target cell. This process requires the collective actions of several viral and cellular proteins, notably, viral glycoproteins B and H (gB and gH). gH is thought to activate the fusogenic function of gB, but how the two proteins interact is unclear. Here, using mutational analysis, we have identified two new functional elements within the cytoplasmic regions of gB and gH: a surface pocket in gB and a single residue in gH, both of which are important for membrane fusion. Based on structural modeling, we propose that the gB pocket is the binding

(https://www.nih.gov/) and by Faculty Scholar grant 55108533 from the Howard Hughes Medical Institute (https://www.hhmi.org/). ZP was supported by grants F30AI161795, T32GM731042 and T32GM731043 from the National Institutes of Health (https://www.nih.gov/). The funders had no role in study design, data collection and analysis, decision to publish, or preparation of the manuscript.

**Competing interests:** The authors have declared that no competing interests exist.

site for the gH residue, and that their interaction activates gB to cause membrane fusion. These findings extend our knowledge of the HSV-1 membrane fusion mechanism. Mechanistic understanding of HSV-1 entry is essential for understanding its pathogenesis and developing new strategies to prevent infections.

## Introduction

Membrane fusion during the entry of enveloped viruses is carried out by viral fusogens, which are proteins displayed on the viral surface that bring the opposing viral and host membranes so close that they merge. To do so, these proteins must refold from the high-energy prefusion conformation into the low-energy postfusion conformation. The energy released upon refolding is thought to overcome the large kinetic barrier associated with membrane fusion (reviewed in [1]). To ensure proper spatial and temporal deployment of viral fusogens, their activity is regulated by environmental signals, such as proton concentration, or interactions with other viral and cellular proteins.

Some of the most complex membrane fusion mechanisms are found in herpesviruses–a family of double-stranded-DNA, enveloped viruses that infect most animal species for life. Their entry requires, at a minimum, three conserved glycoproteins, gB, gH, and gL. gB is a transmembrane glycoprotein composed of an ectodomain, a transmembrane helix, and a cytoplasmic domain that functions as a membrane fusogen. By analogy with other viral fusogens, the refolding of gB from the prefusion to the postfusion conformation is thought to provide the energy for membrane fusion. Indeed, the structures of the prefusion [2,3] and the postfusion forms [4–8] of gB from several herpesviruses suggest large conformational changes that accompany refolding.

gB is a class III fusogen, along with the Vesicular Stomatitis Virus G, baculovirus gp64, and thogotovirus Gp (reviewed in [9]). Like other class III fusogens, gB exists as a trimer. Yet, uniquely, gB is not a stand-alone fusogen activated by exposure to low pH. Instead, gB must be activated by the conserved heterodimeric complex composed of two viral glycoproteins, gH and gL. gH is a transmembrane glycoprotein composed of an ectodomain, a transmembrane helix, and a short cytoplasmic tail. gL is a soluble glycoprotein that binds gH and is required for its proper folding, trafficking to the cell surface, and function [10,11].

The gH/gL heterodimer occupies a central place in the herpesvirus entry and membrane fusion processes because it interacts with several key participants. On the one hand, gH/gL either interacts with the host cell receptors directly or engages viral receptor-binding accessory proteins, depending on the herpesvirus (reviewed in [12, 13]). On the other hand, it binds and activates gB, the fusogen (reviewed in [12,13]). According to the prevalent model [14], interaction with the host cell receptor triggers a cascade of events in which gH/gL transmits the activating signal from the host cell receptor to gB. For example, in the prototypical herpesvirus herpes simplex virus 1 (HSV-1) (reviewed in [15]), which establishes lifelong infections in over a half of people ([16] and reviewed in [17]) and causes oral or genital sores (reviewed in [18]) as well as encephalitis (reviewed in [19–21]), binding of the viral receptor-binding protein, gD, to one of its cognate cellular receptors–nectin-1, herpesvirus entry mediator (HVEM), or 3-OS-modified heparan sulfate ([22, 23] and reviewed in [24])–causes conformational changes in gD that enable it to activate the gH/gL complex [25,26] that, in turn, activates the fusogenic activity of gB [2,5,14,27] (**Fig 1**). But how gB is activated by gH/gL is unknown.

While the gB ectodomain undergoes refolding and interacts with the target cell membrane [30], the cytoplasmic domain of gB (gB$_{CTD}$) is also important for fusion, just as the cytoplasmic

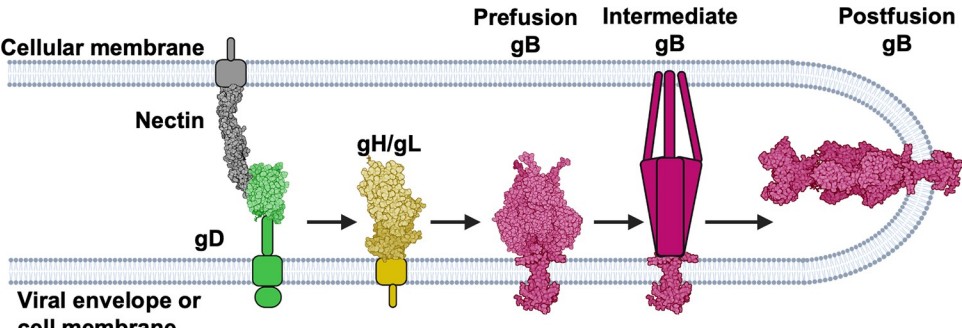

**Fig 1. HSV-1 fusion pathway model.** gD (2C36 [28]) binds a receptor (3U83 [29]) on the target cell and activates gH/gL; gH/gL (3M1C [27]) triggers gB (6Z9M [2] and 5V2S [5]) to refold and cause fusion. gD has been suggested to a dimer [28] but is shown here as a monomer for clarity. Figure created with BioRender.com.

domains of fusogens in viruses such as the Human Immunodeficiency Virus [31] and para-myxoviruses [32]. The $gB_{CTD}$ is thought to inhibit the fusogenic activity of gB because most known $gB_{CTD}$ mutations–C-terminal truncations, point mutations, and insertions–are hyper-fusogenic, i.e., they increase fusion [33–47]. The crystal structure of the full-length HSV-1 gB [5] revealed that the $gB_{CTD}$ is a trimer stabilized by multiple protein/protein and protein/membrane interactions, and that the majority of the hyperfusogenic $gB_{CTD}$ mutations would be predicted to disrupt these stabilizing interactions [5]. Therefore, we have previously proposed that the $gB_{CTD}$ acts as an inhibitory clamp that stabilizes gB in its prefusion form by restricting conformational rearrangements of the gB ectodomain.

In addition to $gB_{CTD}$, the cytoplasmic tail of gH ($gH_{CT}$) is also important for fusion, but instead of inhibiting fusion, it activates it. $gH_{CT}$ truncations decrease fusion in a manner proportional to the length of the truncated sequence [44] such that the shorter the remaining $gH_{CT}$ length, the lower the fusion. Given the apparent inhibitory function of the $gB_{CTD}$ and the activating function of the $gH_{CT}$, previously, we hypothesized that gH may activate gB by using $gH_{CT}$ to bind the inhibitory $gB_{CTD}$ clamp and disrupt it in a wedge-like manner [5]. However, the respective binding sites on $gB_{CTD}$ and $gH_{CT}$ are unknown.

Here, we identified a previously uncharacterized functional site important for fusion within the HSV-1 $gB_{CTD}$, composed of a surface pocket between adjacent protomers. Mutations of residues A851 and T814 located at the bottom of a surface pocket reduce fusion whereas mutations of residues lining the pocket rim are hyperfusogenic, which suggests that the pocket and the rim have important yet opposite roles in fusion. Moreover, we identified $gH_{CT}$ residue V831 as the most functionally important residue within the $gH_{CT}$. When the $gH_{CT}$ is modelled as an extended polypeptide, V831 ends up approximately the same distance from the membrane as the $gB_{CTD}$ pocket, making interactions at these respective sites plausible if gH and gB were to come into proximity. We hypothesize that $gH_{CT}$ residue V831 serves as the wedge that inserts into the newly identified $gB_{CTD}$ pocket. The gB pocket is located above the interface between adjacent protomers, which is akin to a "fault line" within the $gB_{CTD}$ trimer, and we hypothesize that insertion of the gH V831 wedge into the pocket serves to push the protomers apart, which releases the inhibitory clamp. This action would destabilize the inhibitory $gB_{CTD}$ clamp, causing it to release its hold on the gB ectodomain. We hypothesize that in this manner, gH activates the fusogenic activity of gB. The proposed gH-gB triggering mechanism extends our understanding of the regulatory cascade that coordinates HSV-1 entry and may inform new therapeutic approaches aiming at blocking HSV-1 glycoprotein interactions. Both gB and gH are conserved across all herpesviruses, and this activation mechanism could be used by

other gB homologs. Our proposed mechanism emphasizes a central role for the cytoplasmic regions in regulating the activity of a viral fusogen.

## Results

### The A851V gB$_{CTD}$ mutant is hypofusogenic

Unlike the very common hyperfusogenic mutations, hypofusogenic gB$_{CTD}$ mutations, i.e., those that decrease fusion, are rare in HSV-1 and HSV-2 and all result in very low surface expression levels, implying a defect in protein folding [34]. Indeed, within the structure of the trimeric HSV-1 gB$_{CTD}$, these mutations map to the hydrophobic core and have been proposed to cause misfolding by eliminating interactions critical for basal trimer stability [5]. No "true" hypofusogenic mutations in the HSV-1 gB$_{CTD}$ that decrease fusion without affecting protein expression have yet been reported. Interestingly, the HSV-1 gB$_{CTD}$ mutant A851V decreased viral entry [33]. Importantly, we found that the A851V mutation had no effect on the cell surface expression of gB (**Figs 2A and S2**) or gH/gL (**Figs S1 and 3A**), as measured by flow cytometry. Therefore, we hypothesized that A851V reduces viral entry by reducing gB fusogenicity.

To test the hypothesis that the gB A851V mutant was hypofusogenic, we tested its fusogenicity directly. The fusogenic properties of viral fusogens are typically characterized by monitoring cell-cell fusion of uninfected cells expressing viral glycoproteins and host receptors. Here, we used a split-luciferase cell-cell fusion assay reported previously [49], in which the effector and the target cells are transfected with the complementary parts of *Renilla* luciferase. Upon fusion of effector cells with the target cells, functional luciferase forms, and the resulting luminescence is used to quantify fusion (**Fig 2B**). This assay can measure not only early and late extent of fusion, but also early and late rate of fusion, and fusion initiation (**Fig 2C**).

We found that A851V mutation reduced not only the extent of fusion, but also early and late fusion rates while delaying the initiation of fusion (**Fig 2D–2I**). The known hyperfusogenic truncation mutant gB868 was used as a control and had increased extent and rate of fusion as well as earlier fusion initiation (**Fig 2D–2I**). The A851V fusion defect manifested within minutes and was sustained over the entire time course, reaching only ~40% of WT gB fusion by 8 hours. The fusion defect at both early and late steps in fusion suggests that A851V mutation impairs an early, rate-limiting step of fusion. Therefore, A851V is a true hypofusogenic mutation–one that impairs fusogenicity despite being properly folded–the first of its kind reported within the HSV-1 gB$_{CTD}$.

### Mutational analysis of the surface pocket containing A851

A851 is located at the bottom of a surface-exposed pocket that contains another residue, T814 (**Fig 3A and 3B**). The outer rim of the pocket is formed by residues L817, K807, N804, R858, A855, and L852 (**Fig 3A**). The pocket is located at the junction of neighboring protomers such that residues N804, K807, T814, and L817 belong to one protomer, and residues A851, L852, A855 and R858 belong to the neighboring protomer (**Fig 3A**). Thus, the gB$_{CTD}$ trimer contains three symmetry-related pockets.

We first investigated the functional significance of the pocket. A larger hydrophobic side chain of valine in the A851V mutant would reduce the size of the pocket (**Fig 3C**). To further probe the role of A851, we mutated it to a leucine, which has a larger hydrophobic side chain than valine and would be expected to reduce the pocket size further (**Fig 3D**). The other pocket residue, T814, was also mutated to a leucine, keeping up with the strategy of introducing a larger hydrophobic side chain (**Fig 3E**). Just as A851V, both A851L and T814L mutations were hypofusogenic (35%, 26%, and 56% of WT gB fusion at 2 hrs, respectively) (**Fig 3F**). T814L

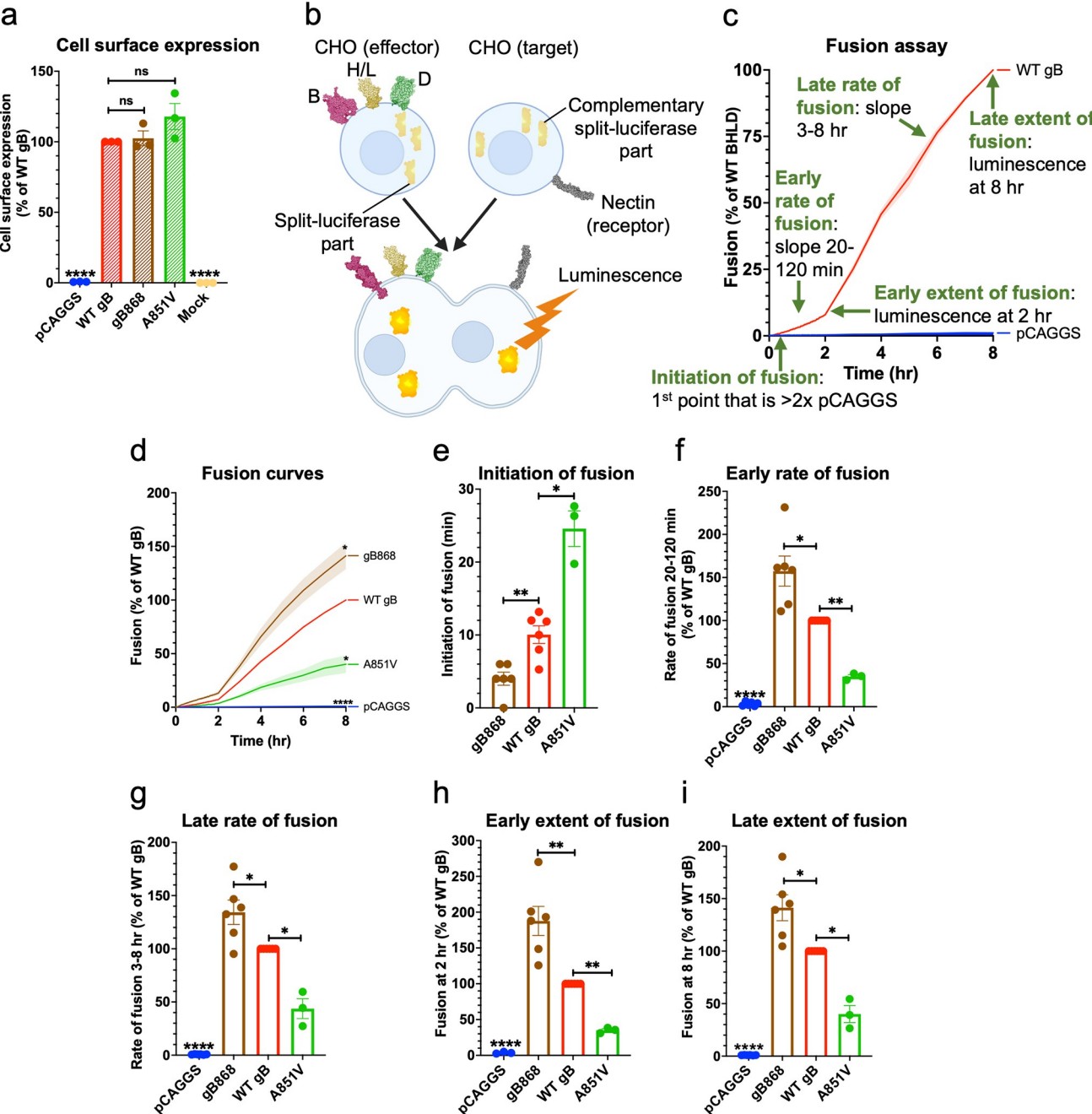

**Fig 2. The gB$_{CTD}$ mutant A851V reduced the rate and extent of fusion. a)** gB A851V and gB868 cell surface expression measured by flow cytometry. R68 primary antibody. Columns show mean. Error bars are SEM. **b)** Split-luciferase cell-cell fusion assay experimental setup. Cells expressing HSV-1 glycoproteins gD (2C36 [28]), gH/gL (3M1C [27]), and gB (6Z9M [2], and 5V2S [5]) fuse with cells expressing a nectin-1 receptor (3U83 [29]). Reconstitution of luciferase reports on fusion. Created with BioRender.com. **c)** Fusion of cells transfected with WT HSV-1 gB, gH, gL, gD compared to pCAGGS. The initiation of fusion is defined as the first reading at which luminescence is greater than twice that of the pCAGGS negative control. Early and late rates of fusion are the slopes of the fusion curves between 20–120 minutes and 3–8 hours post addition of target cells to effector cells, respectively. Early and late extent of fusion is defined as luminescence at 2 and 8 hours post addition of target cells to effector cells, respectively. This represents the total amount of fusion that has occurred over that time. **d)** Fusion of A851V over 8 hr by the split-luciferase fusion assay. *: p < 0.05 compared to WT gB at 8 hr. gB868 was used as a hyperfusogenic positive control [48]. Curve indicates mean values. Shaded area represents SEM. **e-i)** Initiation of fusion, early and late rates and extents of fusion of A851V. Columns show mean. Error bars are SEM. ns: not significant, *: p < 0.05, **: p < 0.01, ****: p <0.0001 in all panels. Data in all panels are from 3–6 independent experiments.

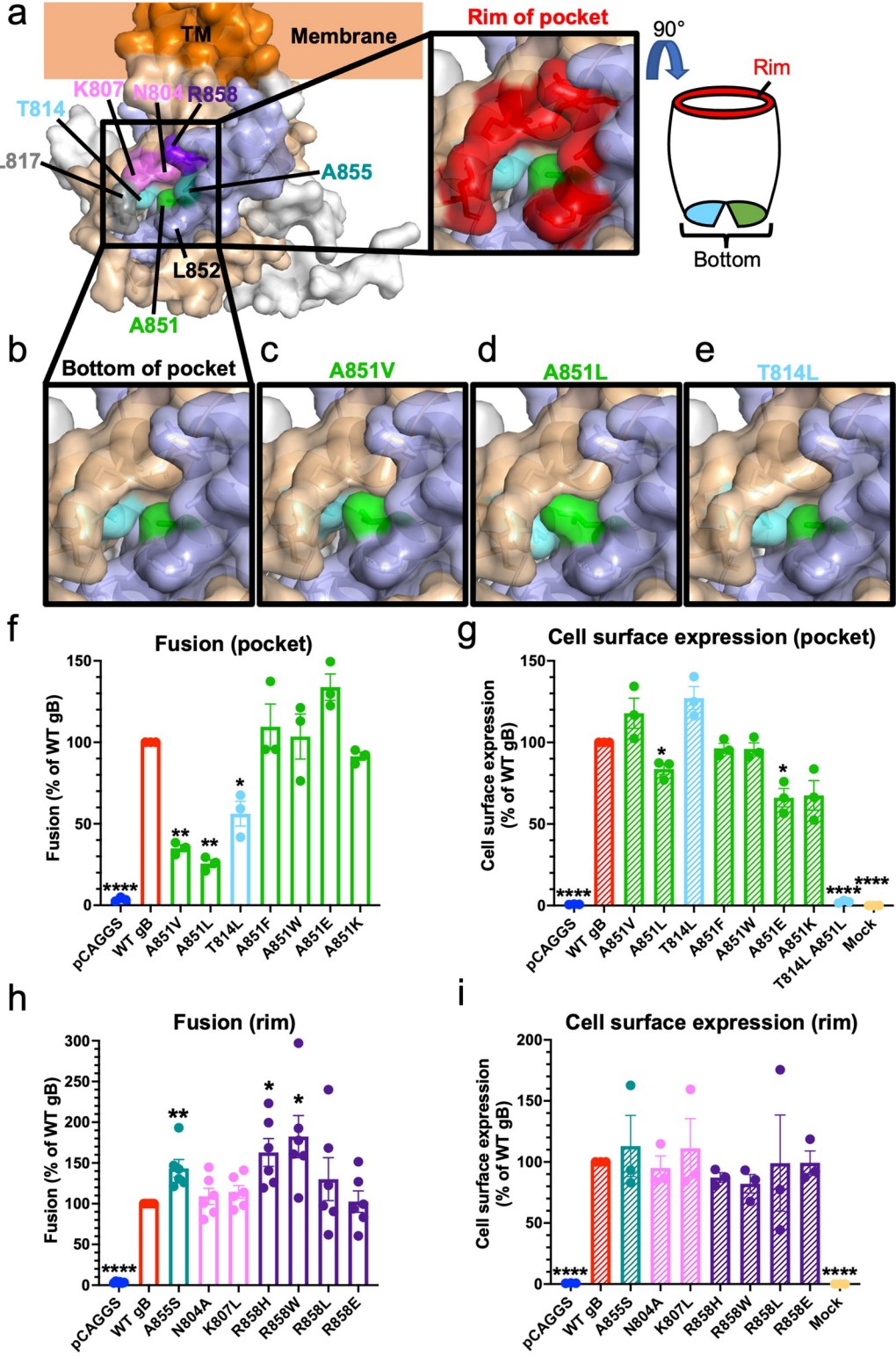

**Fig 3. Fusogenicity and structural effects of mutants in the newly identified gB<sub>CTD</sub> pocket and rim. a)** gB$_{CTD}$ crystal structure and the structure of the pocket on the gB$_{CTD}$. The pocket is formed at the junction of opposing gB$_{CTD}$ protomers, which are colored in wheat and light blue, with the third protomer in white. The residues that form the pocket on the gB$_{CTD}$ are highlighted in colors (left panel). The residues of the outer rim of the pocket are indicated in red (right panel). **b)** The bottom of the pocket is made up of T814 and A851, in sky blue and green, respectively. They do not

contact each other, leaving a space between them at the bottom of the pocket. **c-e)** Pocket mutations of T814 and A851 are hypofusogenic and were modeled in PyMOL. All three hypofusogenic mutations of T814 and A851 were predicted to fill the gB$_{CTD}$ pocket as well as the space between T814 and A851 at the bottom of the pocket. **f)** Fusogenicity of T814 and A851 pocket mutants at 2 hr. A851V data is the same as shown in Fig 2. Fusion trends were the same at 8 hr. **g)** Cell surface expression of pocket mutants by flow cytometry. A851V data is the same as shown in Fig 2. **h)** Fusogenicity of mutations of the pocket rim at 2 hr. Fusion trends were the same at 8 hr. **i)** Cell surface expression of rim mutants. Columns show mean and error bars are SEM in all panels. *: $p < 0.05$, **: $p < 0.01$, ****: $p < 0.0001$ in all panels. Data in all panels are from three to six independent experiments.

had a WT-level of gB cell surface expression whereas the cell surface expression of the A851L mutant was slightly reduced (**Figs 3G and S2**). The mutations had no effect on gH/gL cell surface expression (**Figs S1 and 3A**).

To understand the structural basis of the observed fusion phenotypes, we examined the effect of mutations on the local structure of the gB$_{CTD}$. At the bottom of the pocket, T814 and A851 do not directly contact one another (**Fig 3B**). A851V, A851L, or T814L mutations are all predicted to fill the gB$_{CTD}$ pocket (**Fig 3C–3E**), yet none are expected to destabilize the gB$_{CTD}$ trimer. Given that all three mutations reduce fusion, the pocket appears important for fusion.

To further test this hypothesis, we designed more drastic gB pocket mutations A851F, A851W, and the double mutant T814L/A851L, to occlude the pocket fully, expecting them to decrease fusion to an even greater extent. To investigate the effect of charged residues at this position, we also designed A851E and A851K mutations. The T814L/A851L double mutant was not expressed on the cell surface (**Fig 3G**), possibly, due to protein misfolding caused by a steric clash. A851F and A851W mutants were expressed on the cell surface at WT levels whereas A851E and A851K had slightly reduced cell surface levels (**Figs 3G and S2**). A851F, A851W, and A851K had no effect on gH/gL cell surface expression and A851E had slightly increased gH/gL cell surface expression (**Figs S1 and 3A**). Surprisingly, A851F, A851W, A851K, and A851E had no statistically significant effect on fusion (**Fig 3F**). This was unexpected considering that all substitutions were predicted to completely fill the pocket (**S4A–S4F Fig**).

Further structural modeling revealed that A851F, A851W, and A851K mutations may cause steric clashes with the surrounding residues (**S4G–S4K Fig**), which could destabilize the gB$_{CTD}$ trimer. gB$_{CTD}$ trimer destabilization correlates with a hyperfusogenic phenotype [5]. Therefore, we hypothesized that while filling of the pocket would be expected to reduce fusion, this effect would be counteracted by the hyperfusogenic effect of trimer destabilization, yielding the observed WT fusion levels for each mutant. The charges introduced in A851E and A851K could also be causing a similar counter-effect. Collectively, our findings show that the newly identified surface pocket containing residues A851 and T814 is important for fusion (**Fig 3A**).

## Mutational analysis of the pocket rim

We next mutated residues lining the pocket rim (**Fig 3A**). Mutations were designed to alter side chain polarity, charge, or size (N804A, K807L, A855S, R858W, R858L, R858E) [50] while avoiding large structural changes that could destabilize the gB$_{CTD}$ trimer because such mutations would be expected to have a hyperfusogenic phenotype [5]. Mutations were first modeled in PyMOL [51] and analyzed for significant changes to the surrounding gB$_{CTD}$ structure. Previously reported hyperfusogenic mutations L817H [35] and L817P [36] were not tested here. L852 was not mutated because its mutation L852A was reported to abrogate cell surface expression [34].

Mutations of the rim residues were either hyperfusogenic or had no effect on fusion (**Fig 3H**). All had similar cell surface expression as WT gB (**Figs 3I and S2**) and did not affect gH/gL cell surface expression (**Figs S1 and 3A**). A855S, R858H (also tested in [37,42,44,48]), and R858W were markedly hyperfusogenic. The phenotype of A855S is consistent with the hyperfusogenic phenotype of A855V reported by others [35,40]. The phenotypes of R858H and R858W are consistent with the hyperfusogenic phenotype of R858C reported by others [33]. N804A, K807L, R858L, and R858E had no effect on fusion. According to structural modeling, most mutations are predicted to expose the pocket opening (L817P, N804A, K807L) or the protomeric interface (R858C, R858H, R858L, R858E) whereas some mutations are not (A855S, A855V, and R858W). R858 crosses the protomeric interface between adjacent protomers but lies in the upper rim of the pocket and does not cover the pocket (**Fig 3A**). Therefore, many of its mutations are predicted to expose the protomeric interface without altering the exposure of the pocket opening directly. Many mutations also neutralize a positive charge in the upper portion of the rim (K807L, R858C, R858H, R858W, and R858L). While the predicted structural effects are diverse, the mutations in the rim of the $gB_{CTD}$ pocket are all either hyperfusogenic or fusion-neutral, contrasting with the hypofusogenicity of the pocket mutants. The hyperfusogenic rim mutants are predicted to either expose the pocket or the protomeric interface to a greater extent or neutralize a positive charge, except for A855S and A855V. This suggests a regulatory role for the rim in the fusogenicity of gB.

## gH V831 is the most important $gH_{CT}$ residue for fusion

Mutations that reduce the size of the A851/T814 pocket in $gB_{CTD}$ without destabilizing the trimer–A851V, A851L, and T814L –reduce fusion. Therefore, the size of the pocket is important for fusion, and we hypothesized that it could function as a binding site. Previously, we showed that $gH_{CT}$ has an activating role in fusion because its truncations progressively reduce fusion [44], and proposed that gH may activate gB by using $gH_{CT}$ to disrupt the inhibitory $gB_{CTD}$ trimer in a wedge-like manner [5]. Therefore, we hypothesized that the $gB_{CTD}$ pocket was the binding site for the $gH_{CT}$.

To identify the $gH_{CT}$ residues that could interact with the $gB_{CTD}$ pocket, we first narrowed down residues within the 14-residue $gH_{CT}$ necessary for fusion. In our previous work, we showed that the gH832 truncation mutant lacking 6 residues of the $gH_{CT}$ (**Fig 4A and 4B**) had WT-level fusion whereas the gH829 mutant lacking 9 residues had a significantly reduced fusion [44]. To narrow down residues most important for fusion, we tested serial truncations of gH829, gH830, gH831, and gH832 (**Fig 4B**). gH832 was slightly hyperfusogenic (**Fig 4C**) whereas gH831, gH830, and gH829 were hypofusogenic, with fusion extent proportional to the length of the remaining $gH_{CT}$ (**Fig 4C and 4D**). Cell surface expression of the gH truncation mutants was similar to the WT gH expression (**Figs 4H and S3B**).

To further probe the functional importance of residues T829, S830, V831, and P832, we reversed their polarity [50] by either making them more hydrophobic (T829A, S830A, and P832T) or more hydrophilic (V831T), in the context of the full-length gH (**Fig 4B**). Only V831T was hypofusogenic in a statistically significant manner (**Fig 4E**). There were no significant differences in cell surface expression relative to WT gH, except for P832T, which had slightly increased expression (**Figs 4H and S3**). Collectively, these data implicated V831 as the most important $gH_{CT}$ residue for fusion among those tested.

The $gH_{CT}$ is predicted to be unstructured and contains residues that are disfavored in α-helices, such as proline and valine [52,53]. When the $gH_{CT}$ was modelled as an extended polypeptide, gH V831 ended up approximately the same distance from the membrane as the $gB_{CTD}$ binding pocket (**Fig 5A**). This means that when gH and gB come into proximity, gH

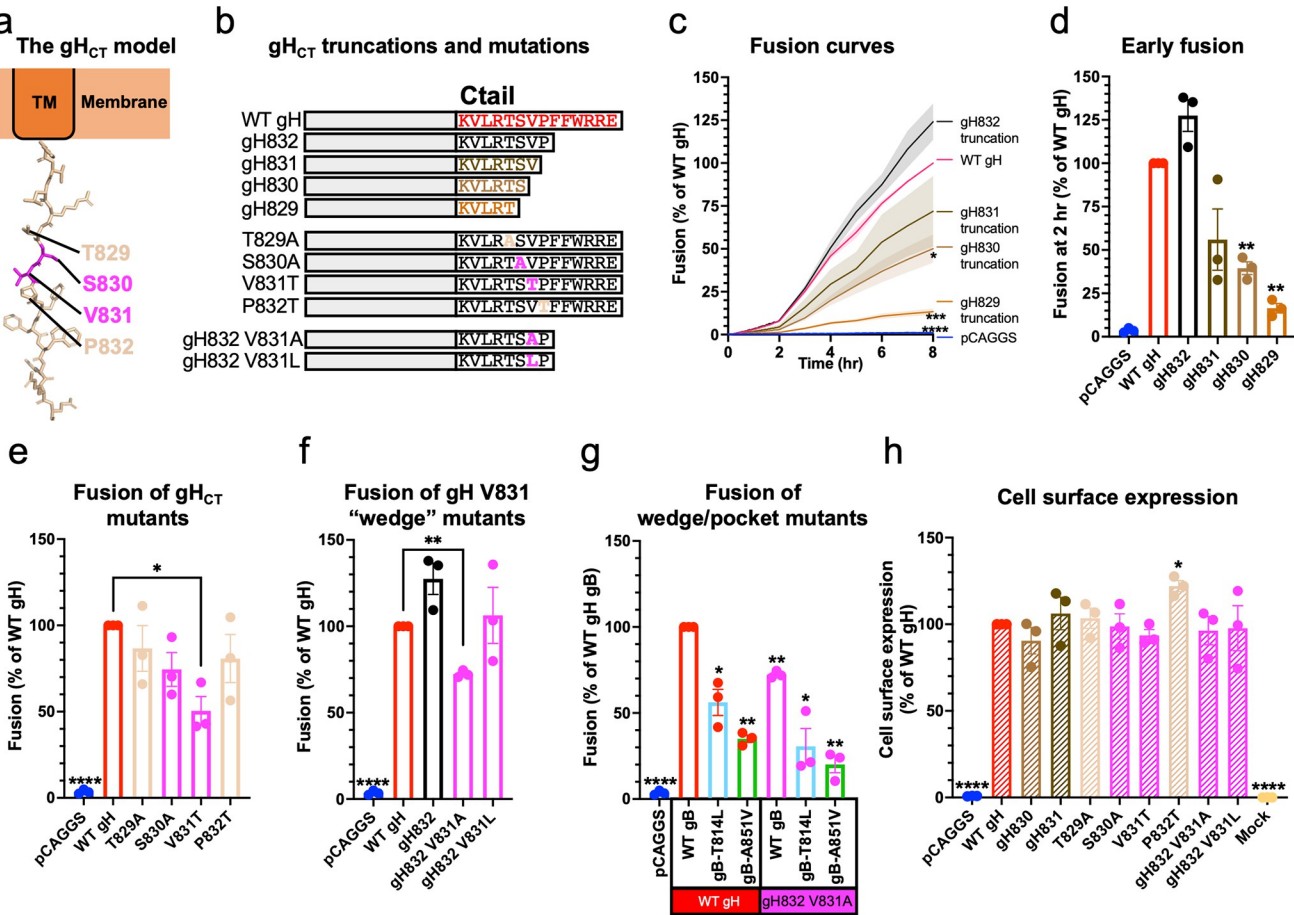

**Fig 4. gH V831 is the most important gH$_{CT}$ residue for fusion. a)** Structural modeling of the gH$_{CT}$. **b)** Summary of gH$_{CT}$ truncations and mutations tested to determine which gH$_{CT}$ residues are the most important for fusion and probe their mechanism of action. **c)** Kinetics of fusion of gH$_{CT}$ truncations over 8 hr. Statistical significance shown is based on comparisons to WT gH fusion at 8 hr. Curves are the mean. Shaded area is SEM. **d)** Fusion of gH$_{CT}$ truncations at 2 hr. **e)** Fusion of mutations of gH829-832 residues at 2 hr to probe the function of the residues. Fusion trends were the same at 8 hr. **f)** Fusion of V831 mutations designed to make the putative gH wedge smaller (V831A gH832) or larger (V831L gH832), at 2 hr. Fusion trends were the same at 8 hr. **g)** Fusion of mutations creating a smaller wedge (V831A gH832) combined with mutations creating smaller pockets (gB T814L, gB A851V), at 2 hr. Fusion trends were the same at 8 hr. **h)** Cell surface expression of the gH$_{CT}$ truncations and mutations. Expression of gH829 and gH832 was determined previously to be the same as WT gH expression [44]. LP11 primary antibody. *: p < 0.05, **: p < 0.01, ***: p < 0.001, ****: p < 0.0001 in all panels. Bars indicate the mean and error bars are SEM. Statistical comparisons are to WT gH and gB. All panels represent averages of three independent experiments.

V831 could, in principle, interact with the gB$_{CTD}$ binding pocket (**Fig 5A**). Therefore, we propose that V831 is the gH$_{CT}$ "wedge" and that it binds the gB$_{CTD}$ pocket containing T814 and A851 (**Fig 5A**).

We next investigated the side chain requirement at the gH residue 831. A hydrophobic side chain appears to be required at this location because threonine could not effectively substitute for valine despite a comparable side chain size (**Fig 4B and 4E**). We then asked whether a smaller (alanine) or a larger (leucine) hydrophobic side chain could support efficient fusion. Previously, we showed that V831A mutation was hypofusogenic [54]. Thus, a smaller hydrophobic side chain of alanine could not substitute for a valine. Interestingly, the hypofusogenic phenotype of V831A was observed only in the context of a truncated gH832 but not full-length gH [54], suggesting that residues 833–838 somehow compensated for the fusion defect of the V831A mutation. To test the effect of a larger hydrophobic side chain at position 831, we

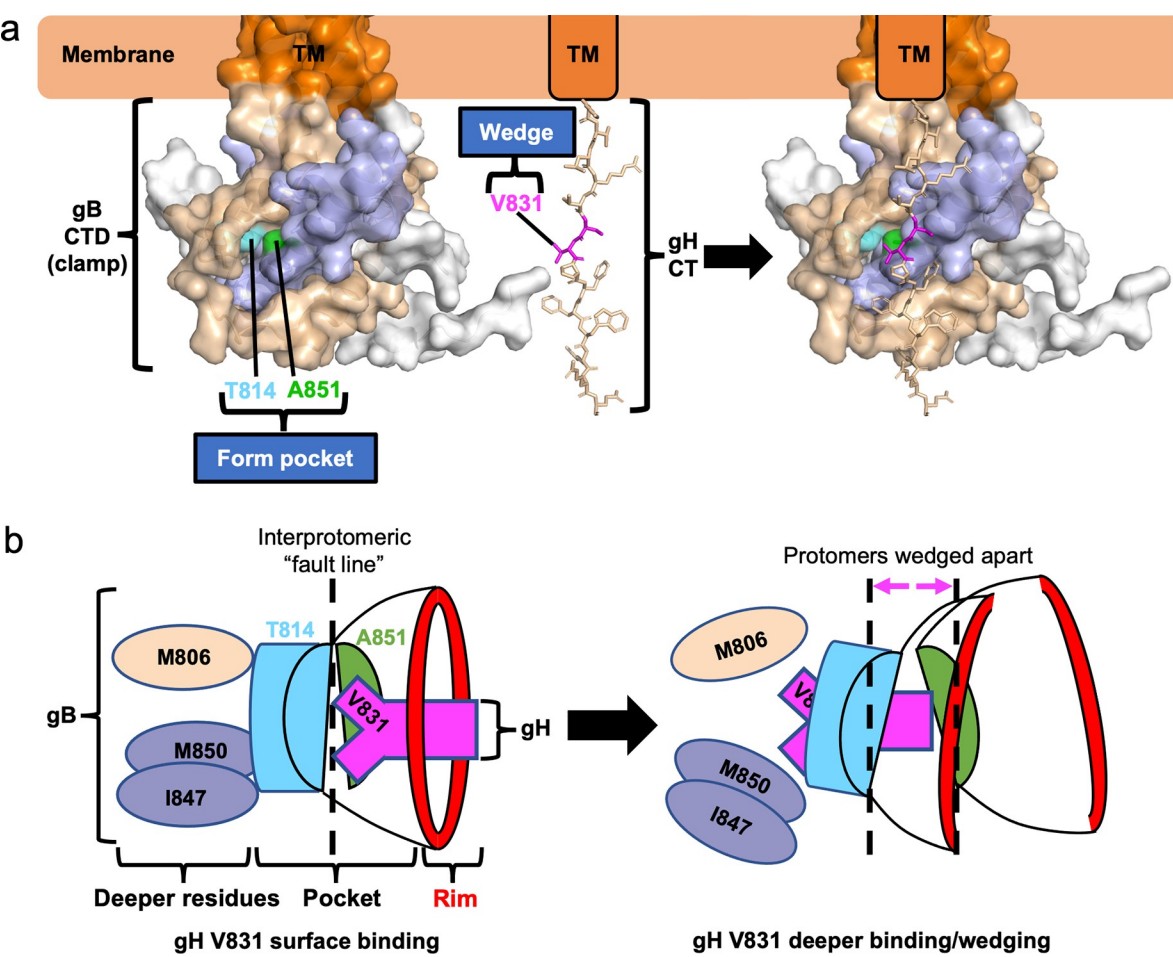

**Fig 5. A model for gB_CTD/gH_CT interactions and fusion triggering. a)** gB_CTD residues T814 (sky blue) and A851 (green) form a gH-binding pocket on the surface of the gB_CTD trimer. gH V831 (magenta) acts as a wedge. Modeling shows that the gH V831 wedge and the gB_CTD pocket are equidistant from the membrane. The V831 wedge binds between gB T814 and A851 in the gB pocket and pushes the gB_CTD protomers (wheat and light blue) apart to destabilize gB and trigger fusogenic refolding of gB into the postfusion conformation. **b)** gH V831 (magenta) acts as a wedge that initially binds between gB_CTD residues T814 (sky blue) and A851 (green) in the pocket on the surface of the gB_CTD trimer. gH V831 then binds to deeper hydrophobic residues of the gB_CTD (colored by protomer), forming favorable hydrophobic interactions. This causes the protomers of gB to be pushed apart as gH V831 enters deeper into the gB_CTD, destabilizing the gB_CTD clamp and triggering gB to refold and cause fusion. The interprotomeric "fault line" refers to the boundary between the wheat and blue protomers, which widens as the protomers are wedged apart.

generated the V831L mutation in the context of the gH832 truncation (gH832 V831L) to prevent potential compensation by residues 833–838 (**Fig 4B**). gH832 V831L had WT fusion level (**Fig 4F**) whereas gH832 V831A was hypofusogenic, in accordance with our previous finding [54]. Neither gH832 V831A nor gH832 V831L mutations had any effect on cell surface expression (**Figs 4H and S3B**). We conclude that valine is minimally required at position 831 to maintain WT levels of fusion.

We then tested the possibility that, in the context of a smaller gB_CTD pocket (gB A851V or T814L), a smaller gH_CT wedge (gH832 V831A) could support fusion by being able to fit into a smaller pocket. We found that the small pocket/small wedge combinations did not restore fusion to WT levels but, instead, reduced fusion in an additive manner (**Fig 4G**). Therefore, we hypothesize that both the gB_CTD pocket and the gH_CT wedge have to be of a certain size to function efficiently.

## Discussion

### Identification of a new functional pocket in the gB$_{CTD}$

Using mutational analysis, we have identified a new functional region of the gB$_{CTD}$ composed of a surface-exposed pocket and its rim (**Fig 3A**). The bottom of the pocket is formed by residues T814 and A851 and the rim is formed by residues N804, K807, L817, L852, A855, and R858. The pocket is located at the junction of two gB protomers, and the pocket and the rim residues are evenly distributed between the neighboring protomers, with N804, K807, T814, and L817 located in one protomer and A851, L852, A855, and R858 in the neighboring protomer. Thus, there are three such pockets within the gB$_{CTD}$ trimer.

Mutations of the gB$_{CTD}$ pocket and rim had opposite effects on fusion. Pocket mutations A851V, A851L, and T814L, which partially filled the pocket, presumably, without otherwise perturbing the surrounding gB$_{CTD}$ structure, significantly decreased fusion, i.e., were hypofusogenic. These three mutations are unusual because, to the best of our knowledge, they are the first hypofusogenic HSV-1 gB$_{CTD}$ mutations that perturbed protein function rather than caused protein misfolding. The hyperfusogenic rim mutations are also unusual because they differed from most other hyperfusogenic mutations in both their location and the presumed mechanism of action. Most hyperfusogenic mutations in the gB$_{CTD}$ [33–45] map to either the protomeric interfaces or the membrane-binding regions and are thus predicted to destabilize the membrane-dependent gB$_{CTD}$ trimer [5]. In contrast, the hyperfusogenic rim mutations, which include mutations of residues L817, A855, and R858 reported here and elsewhere [33, 35–37, 40, 42, 44, 48], are located on the surface and would not be predicted to disrupt the gB$_{CTD}$ trimer. The gB$_{CTD}$ pocket is also located too far from the membrane to participate in membrane interactions. Therefore, the rim mutations increase fusion by a different mechanism, potentially, by exposing the pocket (see below). Collectively, our mutational analysis and structural modeling uncovered a new fusogenic site within the gB$_{CTD}$ trimer composed of two distinct functional regions: the pocket and its rim.

### The gB$_{CTD}$ pocket is a putative binding site for the gH$_{CT}$ wedge, residue V831

Surface pockets often function as binding sites, and mutations that fill surface pockets typically disrupt protein function by blocking binding to protein partners. For example, mutagenesis of a putative chaperone-binding pocket of the p53 protein decreased its expression, indicating that the binding pocket was important for binding the chaperone that stabilizes p53 [55]. Introducing an amino acid with a bulkier, branched side chain into a pocket on Rabies virus P protein resulted in decreased binding by STAT proteins, suggesting that the mutated pocket constitutes the STAT-binding site [56]. Furthermore, in enzymes, active sites, which bind and convert substrates, are typically found in surface pockets, and pocket-filling mutations can either block binding of the substrate to the active site [57] or obstruct its access through a tunnel or channel [58,59].

Mutations that filled the gB$_{CTD}$ pocket decreased fusion, suggesting that the size of the pocket is important for fusion, so we hypothesized that the gB$_{CTD}$ pocket serves as a binding site for a protein partner and that the binding event activates fusion. Previously, we speculated that gH may activate gB by using gH$_{CT}$ to disrupt the inhibitory gB$_{CTD}$ trimer in a wedge-like manner [5]. Therefore, we hypothesized that the newly identified gB$_{CTD}$ pocket was the binding site for the gH$_{CT}$. Using mutational analysis, we identified gH$_{CT}$ residue V831 as the single most critical residue for fusion among those tested. Our structural analysis predicted that if the 14-residue gH$_{CT}$ adopted an extended conformation, gH V831 and the gB$_{CTD}$ pocket would

end up roughly equidistant from the membrane, putting them into ideal positions for reciprocal interaction (**Fig 5A**). Based on these results, we propose that V831 binds in the $gB_{CTD}$ pocket (**Fig 5A**).

We hypothesize that to trigger fusion, the gH V831 side chain binds the $gB_{CTD}$ pocket and inserts deeply enough to push the gB protomers apart like a wedge or a crowbar. This destabilizes the $gB_{CTD}$ clamp, which releases its inhibitory hold on the gB ectodomain, allowing the latter to refold from the prefusion into the postfusion conformation. This hypothesis is supported by the location of the $gB_{CTD}$ pocket right above the interface between adjacent protomers, which is akin to a "fault line" within the $gB_{CTD}$ trimer, a prime location for pushing the protomers apart. Just as the size of the $gB_{CTD}$ pocket is critical for fusion (with a smaller pocket decreasing fusion), so is the size of the sidechain at gH residue 831. A smaller alanine (V831A) decreased fusion whereas a larger leucine (V831L) preserved WT-level fusion. Thus, there appears to be a minimum requirement for the size of the side chain at gH residue 831 for WT-levels of fusion. An alanine would not be able to insert deeply enough between the gB protomers and would be less effective at destabilizing the $gB_{CTD}$, which explains why V831A mutant is hypofusogenic. Interestingly, V831T mutant was also hypofusogenic. The side chain of threonine is similar in size to valine yet is more hydrophilic, which suggests that in addition to size, hydrophobicity of the gH residue 831 is important for fusion.

Implicit in our insertion model is the assumption that the gH V831 residue first interacts with T814 and A851 as it inserts into the pocket and then may interact with nearby residues located within the $gB_{CTD}$ core as it wedges in deeper to push the protomers apart (**Fig 5B**). The residues that lie underneath the $gB_{CTD}$ pocket, M806, I847, and M850, are hydrophobic, so a hydrophobic residue would be a more effective wedge because it could form more favorable interactions with these residues (**Fig 5B**). This helps explain why a slightly larger and hydrophobic leucine (V831L) is an efficient substitute for the native valine whereas a similarly sized yet hydrophilic threonine (V831T) is not. The observation that the residues underneath the $gB_{CTD}$ pocket are hydrophobic further supports our model that the pocket is the binding site for the $gH_{CT}$ wedge, residue V831.

## Role of the rim of the gB pocket in fusion

In contrast to the hypofusogenic mutations of the $gB_{CTD}$ pocket, all mutations of the pocket rim were either hyperfusogenic or fusion-neutral. This included both mutations made in this work (R858W, R858L, R858E, A855S), and those reported previously (R858H [37,42], R858C [33], L817H [35], L817P [36], A855V [35,40]). Being located on the surface, none of these mutations would be predicted to disrupt the $gB_{CTD}$ trimer, in contrast to the majority of the known hyperfusogenic mutations [5]. Therefore, the pocket rim mutations enhance fusion by a different mechanism.

The rim is the entryway into the $gB_{CTD}$ pocket, and some of the hyperfusogenic rim mutations (R858C and R858H) appear to expose the entryway into the pocket to some extent. This may increase fusion by facilitating access of the $gH_{CT}$ wedge to the $gB_{CTD}$ pocket. Some hyperfusogenic rim mutations neutralize the positive charge in the upper portion of the pocket (R858C, R858H, and R858W). Neutralization of a positive charge in the upper portion of the pocket rim could facilitate access of the $gH_{CT}$ wedge to the pocket. Congruent with this idea is that the $gH_{CT}$ is mostly uncharged, notably, residue V831. Some mutants that are predicted to expose the entryway into the pocket (N804A, R858E) or neutralize a positive charge or both (K807L, R858L) were fusion-neutral. It is possible that the effects of these mutations on fusion are more modest than for other mutants, and as a result, statistically significant differences from WT gB could not be detected by our assay.

The A855S and A855V mutations do not change the charge or the access to the pocket. Nonetheless, we hypothesize that by analogy with the other rim mutations listed above, A855S and A855V mutations somehow facilitate $gH_{CT}$ interactions with the $gB_{CTD}$ pocket. The hyperfusogenic phenotype of L817P and L817H mutants is difficult to explain due to insufficient structural data; L817 is the last resolved residue before a disordered loop and its conformation is likely dynamic and difficult to predict.

## More drastic gB A851 pocket mutations may offset the pocket-filling effect with trimer destabilization

Having determined that the $gB_{CTD}$ pocket-reducing mutations A851V, A851L, and T814L all reduced fusion, we had anticipated that more drastic mutations would reduce fusion further. Therefore, we designed mutations A851F and A851W to completely fill the $gB_{CTD}$ pocket and A851E and A851K to both fill the pocket and introduce charge. However, A851F, A851W, A851K, and A851E were fusion-neutral. Structural analysis suggested that due to their large side chains, both phenylalanine and tryptophan would clash with nearby residues causing significant steric strain. Since the $gB_{CTD}$ pocket spans neighboring protomers, we hypothesize that in addition to filling the $gB_{CTD}$ pocket, A851F or A851W also destabilize the $gB_{CTD}$ trimer. Previous mutations predicted to disrupt the $gB_{CTD}$ trimer were hyperfusogenic and were proposed to be more easily triggered by gH/gL due to the $gB_{CTD}$ destabilization, putting the $gB_{CTD}$ on a "hair-trigger" [5]. However, these mutants still required gH/gL, suggesting that the inhibitory effect of the $gB_{CTD}$ clamp in these mutants was weakened but not abrogated. We propose that A851F and A851W have a similar effect. As a result, the hypofusogenic phenotype expected of the pocket-filling mutation is counterbalanced by the hyperfusogenic effect of $gB_{CTD}$ destabilization, resulting in the observed WT-level fusion phenotype. A851K would likewise be predicted to destabilize the $gB_{CTD}$ trimer due to steric strain. Additionally, A851K and A851E could destabilize the $gB_{CTD}$ by the introduction of a charge.

## Potential conservation of gH-gB triggering mechanism in other herpesviruses

The $gB_{CTD}$ sequences are conserved across alphaherpesviruses and, to a lesser extent, among herpesviruses [5]. Therefore, the $gB_{CTD}$ homologs may share a conserved structural fold and the surface pocket observed in HSV-1 $gB_{CTD}$. Indeed, regions containing the pocket and rim residues (HSV-1 gB 804–819 and 849–858) are more highly conserved (**Fig 6A**), with most pocket and rim residues being identical across alphaherpesviruses and similar across herpesviruses examined here (**Fig 6A**). While residue at the position 851 (HSV-1 gB) is less well conserved, three out of six aligned sequences have an alanine (**Fig 6A**). Thus, the sequence analysis supports the hypothesis that the proposed HSV-1 $gH_{CT}$ V831 wedge/$gB_{CTD}$ pocket mechanism of gB triggering by gH could be conserved in other herpesviruses.

On the $gH_{CT}$ side, position 831 (HSV-1 gH) has a conserved valine in HSV-2 and a similar leucine in Epstein-Barr Virus (EBV) (**Fig 6B**). However, in both Varicella Zoster Virus (VZV) and Pseudorabies Virus (PRV), the equivalent position is occupied by serine. A previous study found that HSV-1 gH/gL can trigger PRV gB effectively but not the other way around [61]. This suggests that PRV gH/gL is less effective at activating the fusogenic ability of gB than HSV-1 gH/gL. We hypothesize that the serine in the PRV $gH_{CT}$–and perhaps also in VZV–is a less effective wedge than the valine in HSV-1 gH, similarly to the threonine in the HSV-1 gH V831T mutant. Surprisingly, the 6-residue long $gH_{CT}$ in human cytomegalovirus (HCMV) is much shorter than its counterparts in other herpesviruses. Future studies will examine the gH-gB triggering mechanism across herpesviruses.

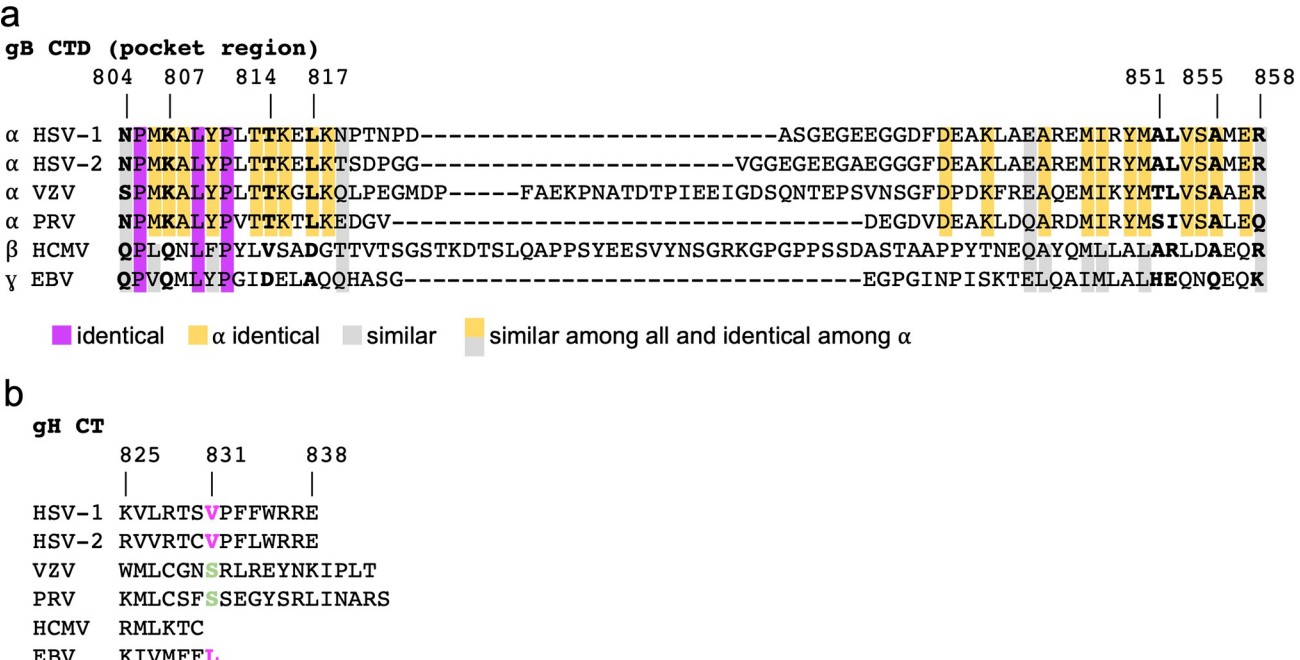

**Fig 6. Conservation of gB_CTD and gH_CT sequences across selected herpesviruses. a)** Alignment of gB_CTD sequences of selected herpesviruses in the region that includes all HSV-1 pocket and rim residues [60]. Chosen sequences include all human alphaherpesviruses (HSV-1, HSV-2, VZV), a closely related non-human alphaherpesvirus (PRV), and important human beta- and gammaherpesviruses (HCMV, EBV). Equivalent residues to the HSV-1 pocket and rim residues are bolded and labeled with HSV-1 residue numbers. Identical and similar residues across herpesviruses are colored. **b)** Alignment of gH_CT sequences of selected herpesviruses [60]. Equivalent residues to the HSV-1 V831 residue is bolded and colored. Pink indicates similar to HSV-1. Green is different from HSV-1.

## Open questions

According to our model of gB triggering, residue V831 within the gH_CT acts as a wedge that binds in the gB_CTD pocket and pushes the gB protomers apart. This destabilizes the gB_CTD trimer, relieving its inhibitory hold on the ectodomain and allowing fusogenic refolding. However, a few unanswered questions remain. First, we do not yet have direct evidence of an interaction between the gB_CTD pocket and the gH_CT wedge. Recently, we reported that gH/gL and gB interact through multiple domains, independently of gD [62]. Disrupting the interactions within the cytoplasmic regions did not reduce overall gH-gB interaction, presumably due to the remaining interactions between the ectodomains and the TMDs. Such extensive interactions between gH and gB present a challenge for future investigations of a direct interaction between the gB_CTD pocket and the gH_CT wedge. Second, it remains to be elucidated what changes occur in gH that cause gH V831 to perform our proposed wedging action in the gB_CTD pocket. We hypothesize that pre-existing interactions between the gB_CTD and the gH_CT [62] position the gH V831 wedge near the gB_CTD pocket. Binding of gD to one of its cognate receptors activates gH/gL, causing it to undergo a conformational change that would push the wedge deeper into the gB_CTD pocket, triggering the fusogenic refolding of gB. Third, it is unknown how the signal from the destabilized gB_CTD trimer is transmitted to the gB ectodomain on the other side of the membrane. A recent study suggested that a conserved regulatory helix in the gB ectodomain proximal to the membrane may be involved in the transduction of the triggering signal from the cytoplasmic domain to the ectodomain [63]. Further studies are needed to decipher how gH/gL interacts with and activates gB and how the gB structure changes in response to this interaction.

## Materials and methods

### Cells and plasmids

CHO cells were received as a gift from J. M. Coffin and grown in Ham's F-12 medium supplemented with 10% fetal bovine serum (FBS), 100 IU penicillin, and 100 μg/ml streptomycin at 37˚ C in the presence of 5% $CO_2$, except as noted otherwise. Plasmids pPEP98, pPEP99, pPEP100, and pPEP101 contain the full-length HSV-1 (strain KOS) gB, gD, gH, and gL genes, respectively, in a pCAGGS vector and were gifts from P. G. Spear [64]. Plasmids RLuc1-7 and RLuc8-11 (carrying the *Renilla* split luciferase genes) and pBG38 (carrying the nectin-1 gene) were gifts from G. H. Cohen and R. J. Eisenberg [49, 65]. Plasmids pJLS11 (gB868) [48], pJLS8 (gB R858H) [48], pJLS15 (gH832) [54], pJLS16 (gH832 V831A) [54], and pHR26 (gH829) [44] were generated previously in our lab.

### gB_{CTD} mutagenesis

Point mutations in the cytoplasmic domain of the full-length gB gene were generated in pPEP98 background by using PCR and Gibson assembly [66]. pPEP98 was cut at PmlI and MfeI sites to generate the backbone for the assembly. Two DNA fragments were created by PCR for the mutants using the primers listed in **S1 Table**. The digested backbone and two DNA inserts generated by PCR were assembled using the Gibson Assembly Master Mix from New England Biolabs. The T814L/A851L double mutant was generated using the same strategy, by making a plasmid with the T814L mutation first and then repeating the steps to introduce the A851L mutation. The A851E mutant was generated by QuikChange PCR [67] using the primers listed in **S1 Table** followed by ligation with T4 ligase. The resulting plasmids were pZP6 (A851V), pZP22 (N804A), pZP23 (K807L), pZP7 (T814L), pZP21 (A851L), pZP25 (A851K), pZP25 (A855S), pZP60 (R858W) pZP24 (R858L), pZP27 (R858E), pZP57 (A851F), pZP58 (A851W), pZP61 (T814L/A851L), pZP2 (A851E).

### gH_{CT} mutagenesis

Point mutations in and truncations of the gH_{CT} were generated in the pPEP100 background by using PCR and Gibson assembly [66]. pPEP100 was cut at MfeI and XhoI sites to generate the backbone for the assembly. Two DNA fragments were created by PCR for the mutants using the primers listed in **S2 Table**. The digested backbone and two DNA inserts generated by PCR were assembled using the Gibson Assembly Master Mix. gH832 V831L was constructed by digesting pJLS15 with MfeI and XhoI to obtain the backbone for the assembly, PCR of the insertion fragment using the primers listed in **S2 Table**, and assembly using the Gibson Assembly Master Mix. The resulting plasmids were pZP32 (gH830), pZP33 (gH831), pZP28 (T829A), pZP29 (S830A), pZP30 (V831T), pZP31 (P832T), pZP62 (gH832 V831L).

### Cell-cell fusion assay

Cell-cell fusion of gB and gH mutants was measured using a split-luciferase assay [49]. Chinese hamster ovary (CHO) cells [68] were seeded into 96-well plates at 50,000 cells per well, three wells per condition, for effector cells and 6-well plates at 200,000 cells per well for target cells. The next day, effector cells were transfected per well with 125 ng gB (pPEP98 or gB mutant) and 41.7 ng each of split luciferase (RLuc1-7), gH (pPEP100 or gH mutant), gL (pPEP101), and gD (pPEP99) using 0.58 μl JetPrime (Polyplus, Illkirch-Graffenstaden, France) in 10 μl JetPrime buffer. For the pCAGGS negative control condition, 250 ng of pCAGGS empty vector was transfected in place of the gB, gH, gL and gD plasmids. CHO cells lack HSV-1 receptors, so no fusion can occur until receptor-bearing target cells are introduced [69]. Each well of target cells was transfected with 1 μg

of the complementary part of the split luciferase (RLuc8-11) and 1 μg of the HSV-1 receptor nectin-1 (pBG38) with 4 μl of JetPrime in 200 μl of JetPrime buffer. On day 3, the tissue culture media was removed from the 96-well plate wells and replaced with 40 μl per well of fusion medium (Ham's F12 with 10% FBS, Penicillin/Streptomycin, 50 mM HEPES), with 1:500 Enduren luciferase substrate (Promega, Madison, WI) added. The Enduren concentration becomes 1:1000 once the target cells are added. Cells were incubated for 1 hr at 37°C. Meanwhile, target cells were detached by incubating with 1 ml per well of Versene (Fisher Scientific, Waltham, MA). Target cells were collected, spun down, and resuspended in 500 μl of fusion medium per well. 40 μl of target cells were added to each well of effector cells. The plate was immediately placed in a BioTek plate reader. Luminescence measurements were taken every 1–2 minutes for 2 hrs followed by measurements every hour until hour 8. Either gB868 or gB R858H was always included as a hyperfusogenic positive control to ensure that the assay was working as expected. The average hyperfusogenic positive control signal was higher than that of the WT condition in all experiments. Luminescence values were then averaged for the three wells in each condition, normalized to the WT signal at 8 hrs, and expressed as a percentage of WT. Reported values are averages of three biological replicates.

## Flow cytometry

Cell surface expression of gB and gH mutants was measured using flow cytometry. CHO cells were seeded at 250,000 cells per well in 6-well plates. The next day, each well was transfected with 2 μg gB (pPEP98 or gB mutant), or pCAGGS, or 1 μg each of gH (pPEP100 or gH mutant) plus 1 μg gL (pPEP101) using 4 μl of JetPrime in 200 μl JetPrime buffer. For conditions testing gH/gL expression in the presence of gB mutants, the transfected DNA was 0.4 μg each of WT gH and gL, and 1.2 μg of gB (WT or mutant). One well per experiment was left untransfected as a 'mock' control. On day 3, the cells were detached with 1 ml per well of Versene and collected using FACS medium (PBS with 3% FBS). Cells were washed with FACS media and incubated for 1 hr on ice with 250 μl of primary antibody (R68 for gB and pCAGGS, LP11 for gH and pCAGGS, anti-c-myc rabbit (A14, Santa Cruz Biotechnology, Dallas, TX) or mouse (9E10, Santa Cruz Biotechnology) antibody as a non-targeting negative control for the Mock condition) at a 1:500 dilution in FACS medium. Cells were washed three times and incubated for 1 hr on ice in the dark with 250 μl secondary FITC-conjugated anti-rabbit antibody (for R68 and rabbit anti-myc; MPBio, Santa Ana, CA) or Alexa Fluor 488 anti-mouse antibody (for LP11 and mouse anti-myc; Invitrogen, Waltham, MA) at a 1:250 dilution in FACS medium. Cells were washed three times and resuspended in 500 μl of FACS medium. Cell fluorescence was measured by flow cytometry. Gating of live cells was performed based on FSC and SSC using FlowJo software. gB+ and gH+ cells were gated using the pCAGGS condition as a negative control, using a cutoff of 5% gB+ or gH + pCAGGS cells to capture the vast majority of true positives while minimizing false positives. Total cell surface expression of the transfected population was obtained by calculating the product of % gB+ or gH+ cells and the mean fluorescence intensity of the gB+ or gH+ cells. Total cell surface expression was then normalized to the WT gB or WT gH condition, expressed as a percentage. The values represent the average of three independent experiments. To generate flow cytometry histograms, the live cells gated using FSC and SSC were used, and their fluorescence was plotted vs cell count. R68 (polyclonal anti-HSV-1 gB) was a gift from G. H. Cohen and R. J. Eisenberg. LP11 (monoclonal anti-HSV-1 gH/gL) was a gift from H. Browne.

## Statistics

Statistical analysis was performed for each experiment on the normalized values using Graph-Pad PRISM 9 software. Unpaired t-test with Welch's correction was used to compare conditions to each other as indicated.

## Structural analysis

The crystal structure of full-length HSV-1 gB, PDB 5V2S [5] was used for *in silico* structural analysis. Predicted effects of $gB_{CTD}$ mutants on the structure were analyzed using PyMOL software ([51] Version 2.5.1 Schrödinger, LLC.). Mutations were introduced one by one. For each rotamer, PyMol provides information about its probability, the degree of structural strain, and the extent of predicted clashing with surrounding residues. The first two parameters are shown as number values whereas the extent of predicted clashing is shown as red disks, with larger disks representing more clashing. Clashing is also apparent on visual inspection based on proximity to surrounding atoms. The rotamers were selected based on a combination of the highest probability, lowest strain, and lowest clashing. Although this strategy should increase the accuracy of modeling, the true rotamer cannot be predicted with high confidence. The area where the mutation was introduced was then "cleaned" in a 5-Å radius using the 'clean' function in PyMOL, which analyzes the selected area and shifts residues into positions that are predicted to be more favorable to minimize energy and strain. The resulting structure after cleaning is more likely to reflect the actual structure of the mutant. The "cleaned" model of the mutant was then visually compared to the WT structure or overlayed onto the WT structure. Predicted changes to salt bridges and H-bonds were also analyzed. The distance of the $gB_{CTD}$ pocket from the membrane is known from the crystal structure of the HSV-1 gB (5V2S [5]). The $gH_{CT}$ was modelled as an unstructured peptide in PyMOL. To compare the distance of gH V831 from the membrane to the distance of the $gB_{CTD}$ pocket from the membrane, the $gH_{CT}$−modelled as an unstructured peptide in PyMol–was lined up such that its first residue (K825) would begin just after the end of the gB transmembrane domain (R796), which represents the border between the membrane and cytoplasm. Visual inspection determined that the positions of the $gB_{CTD}$ pocket and gH V831 coincided and, therefore, were approximately the same distance from the membrane by this analysis. The electrostatic surface potential of the $gH_{CT}$ was calculated using the PyMOL ABPS Tools v. 2.1.5 plugin.

## Supporting information

**S1 Fig. Cell surface expression of WT gH in the presence of gB mutants.** Cell surface expression of gH/gL was tested in the presence of co-transfected gB mutants. LP11 primary antibody. Data are the average of three independent biological replicates.
(TIF)

**S2 Fig. Cell surface expression of gB constructs shown as flow cytometry profiles.** Cell surface expression of gB constructs tested in this work. Mutants tested in separate experiments are shown on separate graphs. The data represent live cells in each condition that were gated using SSC and FSC. FITC signal represents relative levels of gB expression on the cell surface. R68 primary antibody. Data in all panels are from a representative biological replicate.
(TIF)

**S3 Fig. Cell surface expression of gH constructs shown as flow cytometry profiles. a)** Cell surface expression of WT gH/gL in the presence of gB mutants tested in this work. **b)** Cell surface expression of gH/gL constructs tested in this work. Conditions tested in separate experiments are shown on separate graphs. The data represent live cells in each condition that were gated using SSC and FSC. FITC signal represents relative levels of gH/gL expression on the cell surface. LP11 primary antibody. Data in all panels are from a representative biological replicate.
(TIF)

**S4 Fig. Structural effects of more drastic gB A851 pocket mutations. a-b)** gB$_{CTD}$ crystal structure and the structure of the pocket on the gB$_{CTD}$. **c-f)** gB A851F, A851W, A851K, and A851E mutations were modeled in PyMol and are predicted to fill the pocket. **g)** WT gB$_{CTD}$ in cartoon representation to visualize predicted effects of mutations on the surrounding structure. **h-k)** A851F, A851W, A851K, and A851E mutations (green) after energy minimization overlayed onto WT gB (wheat) to show predicted shifts in nearby residues. The large F and W introduced are predicted to cause N804 and K807 to be pushed apart. A851K is predicted to push T814 downwards. A851E is not predicted to cause significant shifts in the nearby residues.
(TIF)

**S1 Table. Primers used for gB$_{CTD}$ mutagenesis.**
(DOCX)

**S2 Table. Primers used for gH$_{CT}$ mutagenesis.**
(DOCX)

## Acknowledgments

We thank Stephen Kwok and Allen Parmelee from the Tufts Flow Cytometry Core and Adam Hilterbrand for flow cytometry training. We thank Roselyn Eisenberg and Gary Cohen (U. Pennsylvania) for the gift of the antibodies and split luciferase plasmids. We thank Doina Atanasiu (U. Pennsylvania) for advice regarding the split luciferase fusion assay. We thank John Coffin, Marta Gaglia, and Karl Munger for helpful discussions and Andrea Rebolledo Viveros for comments on the manuscript. PyMOL software was installed and maintained by SBGrid [70].

## Author Contributions

**Conceptualization:** Zemplen Pataki, Ekaterina E. Heldwein.

**Data curation:** Zemplen Pataki.

**Formal analysis:** Zemplen Pataki.

**Funding acquisition:** Zemplen Pataki, Ekaterina E. Heldwein.

**Investigation:** Zemplen Pataki, Erin K. Sanders.

**Methodology:** Zemplen Pataki.

**Project administration:** Ekaterina E. Heldwein.

**Resources:** Ekaterina E. Heldwein.

**Supervision:** Ekaterina E. Heldwein.

**Validation:** Zemplen Pataki, Ekaterina E. Heldwein.

**Visualization:** Zemplen Pataki.

**Writing – original draft:** Zemplen Pataki, Ekaterina E. Heldwein.

**Writing – review & editing:** Zemplen Pataki, Erin K. Sanders, Ekaterina E. Heldwein.

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
