## [Decision Letter · Decision Letter 0]

11 Apr 2022

Dear Dr. Heldwein,

Thank you very much for submitting your manuscript "A surface pocket in the cytoplasmic domain of the herpes simplex virus fusogen gB controls membrane fusion" for consideration at PLOS Pathogens. As with all papers reviewed by the journal, your manuscript was reviewed by members of the editorial board and by several independent reviewers. In light of the reviews (below this email), we would like to invite the resubmission of a significantly-revised version that takes into account the reviewers' comments.

All three reviewers found the work to be important, interesting and appreciated the significance of the findings.  They recognized the study to be well designed and found the manuscript to be well written.  Important aspects to address in a revised manuscript include addressing concerns about statistical significance, to clarify the impact of transfection efficiency on FACS quantification of HSV cell surface expression, to show gH cell surface expression in Figures 2 and 4 (if possible), to analyze and comment on whether V831 and gBCTD sequences are generally conserved amongst herpesviruses, and to provide additional information about how modeling was done and whether statistical significance can be applied to the modeling.

We cannot make any decision about publication until we have seen the revised manuscript and your response to the reviewers' comments. Your revised manuscript is also likely to be sent to reviewers for further evaluation.

Sincerely,

Benjamin E Gewurz, M.D., Ph.D.

Associate Editor

PLOS Pathogens

Urs Greber

Section Editor

PLOS Pathogens

Kasturi Haldar

Editor-in-Chief

PLOS Pathogens

orcid.org/0000-0001-5065-158X

Michael Malim

Editor-in-Chief

PLOS Pathogens

orcid.org/0000-0002-7699-2064

Reviewer's Responses to Questions

**Part I - Summary**

Reviewer #1: This mutational analysis of HSV entry proteins B and H determined regions of the cytoplasmic tails of these proteins involved in regulating membrane fusion. A panel of mutants were tested in a split-luciferase CHO cell fusion assay to determine fusion activity relative to WT. The level of surface expression was also determined to rule out major effects of mutations on protein folding and surface-targeting. A structural pocket was identified on gBCTD. Mutations of residues inside the pocket tend to reduce fusion, while mutations at the rim of the pocket tend to enhance fusion. On the gH side, residue V831 appears to play a critical role in fusion. Using structural modeling, the authors propose that gH V831 inserts into the gB pocket to act as a wedge and dislocates the gB trimer, thereby allowing conformational changes required for gB fusogenic activity. The model is attractive and consistent with the data. The main weakness of the data resides in the lack of statistical significance of many mutations. Despite this, many non-statistically significant trends of several mutants are heavily incorporated to support the model. The authors acknowledge this issue and the model stands based only on the statistically significant data. This statistical issue seems likely caused by the limited number of experimental repeats (3) for an experiment known to be highly variable in efficiency (transient transfections). The authors must thoroughly revise their manuscript to avoid all overinterpretation, or add to the data for trends to reach statistical significance. The study is well designed and the question addressed is important in the field of herpesvirus entry. The authors also suggest that this gH-gB activation model could be used by other herpesviruses. The relevance of this study would also be strengthened by verifying the conservation of the involved gH residues and gB pocket structure in other viruses. The manuscript is well written and the well-designed experiments have the appropriate controls.

Reviewer #2: In this manuscript, Pataki et al. investigated the mechanism of how HSV-1 gH triggers gB fusogenic activity by using a split-luciferase cell-cell fusion assay. They demonstrated that a surface pocket within the gB CTD and residue V831 within the gH cytoplasmic tail are each important to trigger cell-cell fusion. Overall, this is an important finding which will deepen the knowledge of HSV-1 membrane fusion mechanism. However, the methodology used to perform and evaluated the mutant prediction model is incompletely described, which lowers the confidence of their overall model. They should address following major comments for publication.

Reviewer #3: Herpesvirus entry requires multiple glycoproteins. gH/gL and gB are components of the entry machinery that are conserved in all herpesviruses. gH/gL triggers the viral fusogen gB to mediate fusion; however, the mechanism of gH/gL signaling is unknown. This work addresses this critical gap in knowledge, using structural models and mutagenesis to identify specific sites in both the gB and gH cytoplasmic tails that may interact to trigger gB. The resulting proposed model is that gH tail residue V831 interacts with a pocket in the gB tail (between residues T814 and A851), acting as a wedge that separates the gB protomers and triggers fusion by destabilizing gB. Mutations in gB that obscure the pocket reduce fusion. Mutation of gH-V831 also reduces fusion. If the gH tail is modeled as an extended peptide, gH-V831 lies the same distance from the membrane as this pocket. Unexpected results that some of the gB pocket mutations do not reduce fusion are described forthrightly and the explanation is plausible (that the mutations both obscure the pocket and destabilize the protomer contacts). The work provides a functional explanation for newly and previously identified hyperfusogenic mutations in the gB tail that do not directly disrupt interprotomer contacts (the rim mutants).

The proposed model is clear and supported by carefully considered results. The manuscript is very well written, placing the work in the context of previous studies. The work provides mechanistic explanation of herpesvirus entry, generating a valuable model that is relevant to all herpesviruses and to other viral fusogens.

**Part II – Major Issues: Key Experiments Required for Acceptance**

Reviewer #1: 1) The main issue with the data are the statistics. This led to contradictory statements such as “R858W and R858L were markedly hyperfusogenic, judging by the mean value, even though increased fusion levels of R858W and R858L were not statistically significant, due to large SEM” (page 10, lines 10-12). Transient transfection experiments are notoriously variable in efficacy, resulting in larger SEM. Most experiments are repeated three times, which for such experiments limits the ability of repeated data to reach significance. Even positive hyperfusogenic controls, fail to reach significance (figure 2). It is likely that repeating the experiments a couple of time more may allow more differences to reach statistical significance. To their credit, for the most part, the authors acknowledge this problem, but unfortunately it does not mean statistics can be ignored in global conclusions. This being said, the body of data support the role of the gBCTD pocket in regulating fusion, but stronger statistical significances would certainly help support this model.

2) P6 L13-18: It is surprising than the hyperfusogenic control gB868 is not more significantly different than WT gB. Based on statistics in figure 2, only the early extent of fusion is mildly significantly higher than WT gB. Therefore this sentence in the text is incorrect. However, gB868 has widely been recognised as hyperfusogenic based on less detailed assays.

P10 L21 “the mutations of the rim of gBCTD are overwhelmingly fusogenic with the exception of R858E.” In fact, only two of the seven mutations are significantly hyperfusogenic. This is not overwhelming.

P16 L24: R858E was the only rim mutation that was fusion-neutral. This statement is incorrect since 5 constructs, not only R858E, are not statistically significantly different from WT in fusion assay (figure 3h).

3) Flow cytometry quantification. The quantification of cell surface expression by flow cytometry appears to account for the percentage of transfected cells as well as the level of expression of each transfected cell (product of %transfected and geometric mean of fluorescence). It is unclear what impact the efficiency of transfection has on this quantification. HSV protein surface expression has customarily been tested by cell-based ELISA, which tacitly implies equivalent transfection efficacy of the different plasmids. Do the authors obtain the same significance by simply comparing the mean fluorescence of the SSC/FSC-gated population? Presentation of the FACS profiles would greatly help compare WT gB with mutant gB surface expression.

4) P2 L12-15, P5 L20-22: The authors suggest the interaction of gHCT with the gB pocket described for HSV-1 could be used by other homologs because gB and gH are conserved among herpesviruses. This is interesting but not explored in this study. To maintain this statement in the abstract, it would be worth to analyse specifically the sequences of gHCT homologs to identify if V831 is conserved (or replaced by a residue with similar properties). Similar analysis could be done with gBCTD sequences. In addition, current advances of 3D modeling could also help model gBCTD from other viruses to determine if the pocket structure is maintained. These comparisons would reinforce the relevance of the mechanism presented here for HSV-1.

5) P17 L5: The reason for the apparent absence of phenotype of pocket-filling mutation is that they also push protomers apart, thereby being more likely to separate and be more fusogenic. If that’s the case, wouldn’t we expect these mutants to have exhibit some gH-independent fusion, if these mutations by themselves do what gH V831 is supposed to do? Could the author elaborate in this paragraph?

6) Fig 2a: for completion, gB868 could be included. This is also relevant given the lack of statistical difference in hyperfusogenicity from several parameters detailed in this figure.

Reviewer #2: 1. For Fig 2, please show gH surface expression, to rule out the possibility that gB mutants (eg. A851V) do not disturb gH expression/folding. Same for Fig. 4.

2. Additional information should be provided for how the modeling was done. Please include in the manuscript the prediction models for gB A851F, A851W, and A851K. Please include information on how the accuracy of their predicted structure models was done. Is there a statistical analysis to show that their chosen models has the highest confidence?

3. The evidence provided is not sufficient to support the model presented in figure 5.

-Please describe details on how the distances from the gB CTD pocket or gH V831 to the membrane were estimated.

- Please provide the experimental evidence that the 14aa gHct has an extended conformation.

-Although they have pointed out that they don’t have the direct evidence of the interaction between the gB pocket and the gH V831 interaction, the reviewer still thinks this is a very important piece of data to support their model. Therefore, they should provide direct evidence of gH pocket and gH V831 interaction. They could potentially perform a GST-Pulldown assay or co-IP assay using their mutant set.

- Please provide the experimental evidence that gH V831 side chain can destabilizes the gB CTD inhibitory clamp. They could potentially perform a FRET assay to detect the conformational changes of gB CTD inhibitory clamp upon gH V831 insertion.

Reviewer #3: (No Response)

**Part III – Minor Issues: Editorial and Data Presentation Modifications**

Reviewer #1: 7) P2 L9: the meaning of “fault line” is unclear. The part of the sentence with this jargon may not be necessary in the abstract. However, it should be explained in the text P5 L13. Explanation is provided later (P10 L17). In the abstract, “fault line” could be replaced by protomeric interface.

8) P3 L19: spell out vesicular stomatitis virus.

9) P3 L28-29: This sentence needs to be rephrased for clarity. Indirect interactions with receptor is a bit confusing. It also suggest that gH function is to bind receptors.

10) P11 L17-18: this is slightly overinterpreted. This amino acid is the most important among those tested. Same comment in page 14, line 19.

11) P13 L19: Since the structure is based on a model: “without otherwise presumably perturbing the surrounding gBCTD structure”

12) Figure 1: it should be mentioned that glycoprotein structures are not drawn to scale (a gD monomer is much smaller compared to gH or gB)

13) Legend figure 2 (line 3): PDB numbers and citations are not relevant in this legend and should be replaced by glycoprotein names. “an HSV receptor (3U83 [29])” should be “nectin-1”

14) Figure 3h,i: any significance of the colours of the bars (mostly pink vs violet)

15) Figure 4g: the colour of some value dots (pink) do not match the colour of the bars (blue and green). It may be on purpose, if so, then the left part of the panel should have red dots.

16) Reference 13: typographic error in title

Reviewer #2: 1. Please comment on whether gH V831 and gB T814/A851 are conserved in homologous gH or gB of other herpesviruses?

2. Please remove description of results from the figure legends and instead just focus on key details required to understand each figure.

Reviewer #3: Page 1, line 3: Consider removing the word "unusually" to describe control of fusion by the cytoplasmic tail of the fusion protein. HIV and paramyxovirus fusion proteins are also influenced by their cytoplasmic tails.

Page 6, line 19: Clarify what "impairs fusion rather than refolding" means. Does this statement mean simply that A851V was not misfolded or does it mean that the mutant does not affect gB refolding? I believe this sentence means that A851V is the first properly folded hypofusogenic gBCTD mutant, but I wasn't sure.

Page 10, line 16 and Figure 5b: Clarify the concept of the "fault line". Does "fault line" simply mean the interface between adjacent protomers near this pocket that is disrupted to trigger fusion?

Fig. 4e: Does the horizontal line for statistical comparison apply only to V831T? If so, consider drawing the line as in Fig. 4f, where the vertical lines do not cross the horizontal lines. The current Fig. 4e could be interpreted to mean that all the mutants under the line have p <0.05.

Brief additions to the discussion:

If the authors know whether the gB hyperfusogenic mutants (the new A851 mutants or relevant previously characterized mutants) retain gH/gL-dependence for fusion, they could discuss why mutations in the gB pocket or rim alone are insufficient to trigger gB. This is not a request for additional experiments, as the findings are not critical for the conclusions of this work.

In the 'open questions' section, the authors could comment on whether they expect that the ectodomains of gB and gH/gL also interact (in addition to the tails).

In the 'open questions' section, the authors could comment on how they anticipate that gH-V831 regulates fusion, since gH-V831 does not act as a wedge for gB until signaled by a receptor-binding protein interacting with gH/gL (presumably in the ectodomain).

PLOS authors have the option to publish the peer review history of their article (what does this mean?). If published, this will include your full peer review and any attached files.

Reviewer #1: No

Reviewer #2: No

Reviewer #3: No
---

## [Decision Letter · Decision Letter 1]

3 Jun 2022

Dear Dr. Heldwein,

We are pleased to inform you that your manuscript 'A surface pocket in the cytoplasmic domain of the herpes simplex virus fusogen gB controls membrane fusion' has been provisionally accepted for publication in PLOS Pathogens.

Best regards,

Benjamin E Gewurz, M.D., Ph.D.

Associate Editor

PLOS Pathogens

Urs Greber

Section Editor

PLOS Pathogens

Kasturi Haldar

Editor-in-Chief

PLOS Pathogens

orcid.org/0000-0001-5065-158X

Michael Malim

Editor-in-Chief

PLOS Pathogens

orcid.org/0000-0002-7699-2064

Reviewer Comments (if any, and for reference):

Reviewer's Responses to Questions

**Part I - Summary**

Reviewer #1: In their revised manuscripts, the authors carefully addressed the reviewers comments. In particular, they repeated experiments to strengthen the statistical significance of the data. These data are now supporting the proposed model in a more robust manner. They also reanalyzed expression data and added supplementary data (FACS profiles) that alleviate potential technical issues on protein expression as the result of transfection efficacy. To address questions relative to the direct interactions between gB and gHgL, the authors refer, and provide the BioRxiv link, to a pre-peer review manuscript. To improve the discussion, the authors added figure 6 to address the commonality of the proposed mechanism across human herpesviruses. Finally, more technical details have been added to clarify the modeling approach. Overall, these careful modification strengthen the manuscript.

Reviewer #2: All the comments have been addressed.

Reviewer #3: This work provides a mechanistic explanation of herpesvirus entry, generating a valuable model that is relevant to all herpesviruses. The discussion section of the paper places the work in context with the field, providing compelling hypotheses about the triggering of other herpesviruses and highlighting the remaining open questions.

The authors thoroughly responded to the previous reviewers comments, including increasing replicates for fusion experiments to refine statistical significance, performing additional experiments to determine the impact of gB mutant coexpression on gHgL surface expression, including details of gB surface expression flow cytometry analysis, and adding an alignment of gH and gB tails.

The only requested experiments that the authors did not perform were (1) a coIP or GST-pulldown assay to demonstrate that gH V831 interacts with the gB-CTD pocket and (2) FRET studies to show that gH V831 destabilizes the gB-CTD. I agree with the author’s reasoning that a pull-down experiment would be unlikely to demonstrate a direct interaction between gH V831 and the gB-CTD pocket because gHgL and gB may interact at multiple sites. I also agree that develop a FRET assay to demonstrate gH-gB tail interactions is outside the scope of this publication.

**Part II – Major Issues: Key Experiments Required for Acceptance**

Reviewer #1: None noted

Reviewer #2: (No Response)

Reviewer #3: (No Response)

**Part III – Minor Issues: Editorial and Data Presentation Modifications**

Reviewer #1: None noted

Reviewer #2: (No Response)

Reviewer #3: (No Response)

PLOS authors have the option to publish the peer review history of their article (what does this mean?). If published, this will include your full peer review and any attached files.

Reviewer #1: No

Reviewer #2: No

Reviewer #3: No

---

## [Editor Report · Acceptance letter]

16 Jun 2022

Dear Dr. Heldwein,

We are delighted to inform you that your manuscript, "A surface pocket in the cytoplasmic domain of the herpes simplex virus fusogen gB controls membrane fusion," has been formally accepted for publication in PLOS Pathogens.

Best regards,

Kasturi Haldar

Editor-in-Chief

PLOS Pathogens

orcid.org/0000-0001-5065-158X

Michael Malim

Editor-in-Chief

PLOS Pathogens

orcid.org/0000-0002-7699-2064